# Lower Bounds on Adversarial Robustness from Optimal Transport

**Arjun Nitin Bhagoji** [*]
Department of Electrical Engineering
Princeton University
abhagoji@princeton.edu

**Daniel Cullina** [*,†]
Department of Electrical Engineering
Pennsylvania State University
cullina@psu.edu

**Prateek Mittal**
Department of Electrical Engineering
Princeton University
pmittal@princeton.edu

## Abstract

While progress has been made in understanding the robustness of machine learning classifiers to test-time adversaries (evasion attacks), fundamental questions remain unresolved. In this paper, we use optimal transport to characterize the minimum possible loss in an adversarial classification scenario. In this setting, an adversary receives a random labeled example from one of two classes, perturbs the example subject to a neighborhood constraint, and presents the modified example to the classifier. We define an appropriate cost function such that the minimum transportation cost between the distributions of the two classes determines the *minimum* $0-1$ *loss for any classifier*. When the classifier comes from a restricted hypothesis class, the optimal transportation cost provides a lower bound. We apply our framework to the case of Gaussian data with norm-bounded adversaries and explicitly show matching bounds for the classification and transport problems as well as the optimality of linear classifiers. We also characterize the sample complexity of learning in this setting, deriving and extending previously known results as a special case. Finally, we use our framework to study the gap between the optimal classification performance possible and that currently achieved by state-of-the-art robustly trained neural networks for datasets of interest, namely, MNIST, Fashion MNIST and CIFAR-10.

## 1   Introduction

Machine learning (ML) has become ubiquitous due to its impressive performance in a wide variety of domains such as image recognition [48,72], natural language and speech processing [22,25,37], game-playing [12,59,71] and aircraft collision avoidance [42]. This ubiquity, however, provides adversaries with both the opportunity and incentive to strategically fool machine learning systems during both the training (poisoning attacks) [5,9,40,60,67] and test (evasion attacks) [8,17,34,57,58,63,77] phases. In an *evasion attack*, an adversary adds imperceptible perturbations to inputs in the test phase to cause misclassification. A large number of adversarial example-based evasion attacks have been proposed against ML algorithms used for tasks such as image classification [8,17,19,34,63,77], object detection [21,53,83], image segmentation [2,31] and speech recognition [18,86]; generative

---

[*]Equal contribution.
[†]Work done while at Princeton University

models for image data [45] and even reinforcement learning algorithms [38, 46]. These attacks have been carried out in black-box [7, 11, 20, 52, 61, 62, 77] as well as in physical settings [29, 49, 70, 74].

A wide variety of defenses based on adversarial training [34, 54, 78], input de-noising through transformations [6, 24, 28, 69, 84], distillation [65], ensembling [1, 4, 75] and feature nullification [81] were proposed to defend ML algorithms against evasion attacks, only for most to be rendered ineffective by stronger attacks [3, 14–16]. Iterative adversarial training [54] is a current state-of-the-art empirical defense. Recently, defenses that rely on adversarial training and are provably robust to small perturbations have been proposed [35, 44, 66, 73] but are unable to achieve good generalization behavior on standard datasets such as CIFAR-10 [47]. In spite of an active line of research that has worked to characterize the difficulty of learning in the presence of evasion adversaries by analyzing the sample complexity of learning classifiers for known distributions [68] as well as in the distribution-free setting [23, 56, 85], fundamental questions remain unresolved. One such question is, *what is the behavior of the optimal achievable loss in the presence of an adversary?*

In this paper, we derive bounds on the $0 - 1$ loss of classifiers while classifying adversarially modified data at test time, which is often referred to as *adversarial robustness*. We first develop a framework that relates classification in the presence of an adversary and optimal transport with an appropriately defined adversarial cost function. For an arbitrary data distribution with two classes, we characterize *optimal adversarial robustness* in terms of the transportation distance between the classes. When the classifier comes from a restricted hypothesis class, we obtain a lower bound on the minimum possible $0 - 1$ loss (or equivalently, an upper bound on the maximum possible classification accuracy).

We then consider the case of a mixture of two Gaussians and derive matching upper and lower bounds for adversarial robustness by framing it as a convex optimization problem and proving the optimality of linear classifiers. For an $\ell_\infty$ adversary, we also present the explicit solution for this optimization problem and analyze its properties. Further, we derive an expression for sample complexity with the assumption of a Gaussian prior on the mean of the Gaussians which allows us to independently match and extend the results from Schmidt et al. [68] as a special case.

Finally, in our experiments, we find transportation costs between the classes of empirical distributions of interest such as MNIST [50], Fashion-MNIST [82] and CIFAR-10 [47] for adversaries bounded by $\ell_2$ and $\ell_\infty$ distance constraints, and relate them to the classification loss of state-of-the-art robust classifiers. Our results demonstrate that as the adversarial budget increases, the gap between current robust classifiers and the lower bound increases. This effect is especially pronounced for the CIFAR-10 dataset, providing a clear indication of the difficulty of robust classification for this dataset.

**What do these results imply?** First, the effectiveness of any defense for a given dataset can be directly analyzed by comparing its robustness to the lower bound. In particular, this allows us to identify regimes of interest where robust classification is possible. Our bound can be used to decide whether a particular adversarial budget is big or small. Second, since our lower bound does not require any distributional assumptions on the data, we are able to directly apply it to empirical distributions, characterizing whether robust classification is possible.

Further, in the Gaussian setting, the optimal classifier in the adversarial case depends explicitly on the adversary's budget. The optimal classifier in the benign case (corresponding to a budget of $0$), differs from that for non-zero budgets. This immediately *establishes a trade-off* between the benign accuracy and adversarial robustness achievable with a given classifier. This raises interesting questions about which classifier should actually be deployed and how large the trade-off is. From the explicit solution we derive in the Gaussian setting, we observe that non-robust features occur during classification due to a mismatch between the norms used by the adversary and that governing the data distribution. We expand upon this observation in Section 4.1, which was also made independently by Ilyas et al. [39].

**Contributions:** We summarize our contributions in this paper as follows: i) we develop a framework for finding general lower bounds for classification error in the presence of an adversary (adversarial robustness) using optimal transport, ii) we show matching upper and lower bounds for adversarial robustness as well as the sample complexity of attaining it for the case of Gaussian data and a convex, origin-symmetric constraint on the adversary and iii) we determine lower bounds on adversarial robustness for empirical datasets of interest and compare them to those of robustly trained classifiers.

## 2 Preliminaries and Notation

In this section, we set up the problem of learning in the presence of an evasion adversary. Such an adversary presents the learner with adversarially modified examples at test time but does not interfere with the training process [17, 34, 77]. We also define notation for the rest of the paper and explain how other work on adversarial examples fits into our setting.

| Symbol | Usage |
|---|---|
| $\mathcal{X}$ | Space of natural examples |
| $\tilde{\mathcal{X}}$ | Space of examples produced by the adversary |
| $N : \mathcal{X} \to 2^{\tilde{\mathcal{X}}}$ | Neighborhood constraint function for adversary |
| $P$ | Distribution of labeled examples (on $\mathcal{X} \times \{-1, 1\}$) |

Table 1: Basic notation for the adversarial learning problem

We summarize the basic notation in Table 1. We now formally describe the learning problem. There is an unknown $P \in \mathbb{P}(\mathcal{X} \times \{-1, 1\})$. The learner receives labeled training data $(\mathbf{x}, \mathbf{y}) = ((x_0, y_0), \dots, (x_{n-1}, y_{n-1})) \sim P^n$ and must select a hypothesis $h$. The evasion adversary receives a labeled natural example $(x_{\text{Test}}, y_{\text{Test}}) \sim P$ and selects $\tilde{x} \in N(x_{\text{Test}})$, the set of adversarial examples in the neighborhood of $x_{\text{Test}}$. The adversary gives $\tilde{x}$ to the learner and the learner must estimate $y_{\text{Test}}$. Their performance is measured by the 0-1 loss, $\ell(y_{\text{Test}}, h(\tilde{x}))$.

Examples produced by the adversary are elements of a space $\tilde{\mathcal{X}}$. In most applications, $\mathcal{X} = \tilde{\mathcal{X}}$, but we find it useful to distinguish them to clarify some definitions. We require $N(x)$ to be nonempty so some choice of $\tilde{x}$ is always available. By taking $\mathcal{X} = \tilde{\mathcal{X}}$ and $N(x) = \{x\}$, we recover the standard problem of learning without an adversary. If $N_1, N_2$ are neighborhood functions and $N_1(x) \subseteq N_2(x)$ for all $x \in \mathcal{X}$, $N_2$ represents a stronger adversary. When $\mathcal{X} = \tilde{\mathcal{X}}$, a neighborhood function $N$ can be defined using a distance $d$ on $\mathcal{X}$ and an adversarial constraint $\beta$: $N(x) = \{\tilde{x} : d(x, \tilde{x}) \le \beta\}$. This provides an ordered family of adversaries of varying strengths used in previous work [17, 34, 68].

The learner's error rate under the data distribution $P$ with an adversary constrained by the neighborhood function $N$ is $L(N, P, h) = \mathbb{E}_{(x,y) \sim P}[\max_{\tilde{x} \in N(x)} \ell(h(\tilde{x}), y)]$.

## 3 Adversarial Robustness from Optimal transport

In this section, we explain the connections between adversarially robust classification and optimal transport. At a high level, these arise from the following idea: if a pair of examples, one from each class, are adversarially indistinguishable, then any hypothesis can classify at most one of the examples correctly, By finding families of such pairs, one can obtain *lower bounds on classification error rate*. When the set of available hypotheses is as large as possible, the best of these lower bounds is tight.

**Section Roadmap:** We will first review some basic concepts from optimal transport theory [80]. Then, we will define a cost function for adversarial classification as well as its associated potential functions that are needed to establish Kantorovich duality. We show how a coupling between the conditional distributions of the two classes can be obtained by composing couplings derived from the adversarial strategy and the total variation distance, which links hypothesis testing and transportation costs. Finally, we show that the potential functions have an interpretation in terms of classification, which leads to our theorem connecting adversarial robustness to the optimal transport cost.

### 3.1 Basic definitions from optimal transport

In this section, we use capital letters for random variables and lowercase letters for points in spaces.

**Couplings** A coupling between probability distributions $P_X$ on $\mathcal{X}$ and $P_Y$ on $\mathcal{Y}$ is a joint distribution on $\mathcal{X} \times \mathcal{Y}$ with marginals $P_X$ and $P_Y$. Let $\Pi(P_X, P_Y)$ be the set of such couplings.

**Definition 1** (Optimal transport cost). *For a cost function $c : \mathcal{X} \times \mathcal{Y} \to \mathbb{R} \cup \{+\infty\}$ and marginal distributions $P_X$ and $P_Y$, the optimal transport cost is*

$$C(P_X, P_Y) = \inf_{P_{XY} \in \Pi(P_X, P_Y)} \mathbb{E}_{(X,Y) \sim P_{XY}}[c(X, Y)]. \tag{1}$$

**Potential functions and Kantorovich duality**    There is a dual characterization of optimal transport cost in terms of potential functions which we use to make the connection between the transport and classification problems.

**Definition 2** (Potential functions). *Functions $f : \mathcal{X} \to \mathbb{R}$ and $g : \mathcal{Y} \to \mathbb{R}$ are potential functions for the cost $c$ if $g(y) - f(x) \leq c(x, y)$ for all $(x, y) \in \mathcal{X} \times \mathcal{Y}$.*

A pair of potential functions provide a one-dimensional representation of the spaces $\mathcal{X}$ and $\mathcal{Y}$. This representation must be be faithful to the cost structure on the original spaces: if a pair of points $(x, y)$ are close in transportation cost, then $f(x)$ must be close to $g(y)$. In the dual optimization problem for optimal transport cost, we search for a representation that separates $P_X$ from $P_Y$ as much as possible:

$$C(P_X, P_Y) = \sup_{f,g} \mathbb{E}_{Y \sim P_Y}[g(Y)] - \mathbb{E}_{X \sim P_X}[f(X)]. \tag{2}$$

For any choices of $f$, $g$, and $P_{XY}$, it is clear that $\mathbb{E}[g(Y)] - \mathbb{E}[f(X)] \leq \mathbb{E}[c(X, Y)]$. Kantorovich duality states that there are in fact choices for $f$ and $g$ that attain equality.

Define the dual of $f$ relative to $c$ to be $f^c(y) = \inf_x c(x, y) + f(x)$. This is the largest function that forms a potential for $c$ when paired with with $f$. In (2), it is sufficient to optimize over pairs $(f, f^c)$.

**Compositions**    The composition of cost functions $c : \mathcal{X} \times \mathcal{Y} \to \mathbb{R}$ and $c' : \mathcal{Y} \times \mathcal{Z} \to \mathbb{R}$ is
$$(c \circ c') : \mathcal{X} \times \mathcal{Z} \to \mathbb{R} \qquad (c \circ c')(x, z) = \inf_{y \in \mathcal{Y}} c(x, y) + c'(y, z).$$

The composition of optimal transport costs can be defined in two equivalent ways:
$$(C \circ C')(P_X, P_Z) = \inf_{P_Y} C(P_X, P_Y) + C'(P_Y, P_Z) = \inf_{P_{XZ}} \mathbb{E}[(c \circ c')(X, Z)]$$

**Total variation distance**    The total variation distance between distributions $P$ and $Q$ is
$$C_{\text{TV}}(P, Q) = \sup_A P(A) - Q(A). \tag{3}$$

We use this notation because it is the optimal transport cost for the cost function $c_{\text{TV}} : \mathcal{X} \times \mathcal{X} \to \mathbb{R}$, $c_{\text{TV}}(x, x') = \mathbf{1}[x \neq x']$.   Observe that (3) is equivalent to (2) with the additional restrictions that $f(x) \in \{0, 1\}$ for all $x$, i.e. $f$ is an indicator function for some set $A$ and $g = f^{c_{\text{TV}}}$.

For binary classification with a symmetric prior on the classes, a set $A$ that achieves the optimum in Eq. (3) corresponds to an optimal test for distinguishing $P$ from $Q$.

## 3.2    Adversarial cost functions and couplings

We now construct specialized version of costs and couplings that translate between robust classification and optimal transport.

**Cost functions for adversarial classification**    The adversarial constraint information $N$ can be encoded into the following cost function $c_N : \mathcal{X} \times \tilde{\mathcal{X}} \to \mathbb{R}$: $c_N(x, \tilde{x}) = \mathbf{1}[\tilde{x} \notin N(x)]$.   The composition of $c_N$ and $c_N^\top$ (i.e. $c_N$ with the arguments flipped) has simple combinatorial interpretation: $(c_N \circ c_N^\top)(x, x') = \mathbf{1}[N(x) \cap N(x') = \varnothing]$.

Perhaps the most well-known example of optimal transport is the earth-mover's or 1-Wasserstein distance, where the cost function is a metric on the underlying space. In general, the transportation cost $c_N \circ c_N^\top$ is not a metric on $\mathcal{X}$ because $(c_N \circ c_N^\top)(x, x') = 0$ does not necessarily imply $x = x'$. However, when $(c_N \circ c_N^\top)(x, x') = 0$, we say that the points are *adversarially indistinguishible*.

**Couplings from adversarial strategies**    Let $a : \mathcal{X} \to \tilde{\mathcal{X}}$ be a function such that $a(x) \in N(x)$ for all $x \in \mathcal{X}$.   Then $a$ is an admissible adversarial perturbation strategy.   The adversarial expected risk can be expressed as a maximization over adversarial strategies: $L(N, P, h) = \sup_{a_1, a_{-1}} \mathbb{E}_{(x,c) \sim P}[\ell(h(a_c(x)), c)]$. Let $\tilde{X}_1 = a_1(X_1)$, so $a_1$ gives a coupling $P_{X_1 \tilde{X}_1}$ between $P_{X_1}$ and $P_{\tilde{X}_1}$. By construction, $C_N(P_{X_1}, P_{\tilde{X}_1}) = 0$. A general coupling between $P_{X_1}$ and $P_{\tilde{X}_1}$ with $C_N(P_{X_1}, P_{\tilde{X}_1}) = 0$ corresponds to a randomized adversarial strategy.

We define $P_{\tilde{X}_{-1}}$ and $P_{X_{-1}\tilde{X}_{-1}}$ analogously. By composing the adversarial strategy coupling $P_{X_1 \tilde{X}_1}$, the total variation coupling of $P_{\tilde{X}_1}$ and $P_{\tilde{X}_{-1}}$, and $P_{\tilde{X}_{-1}X_{-1}}$, we obtain a coupling $P_{X_1 X_{-1}}$.

**Potential functions from classifiers** Now we can explore the relationship between transport and classification. Consider a given hypothesis $h : \tilde{\mathcal{X}} \to \{-1, 1\}$. A labeled adversarial example $(\tilde{x}, y)$ is classified correctly if $\tilde{x} \in h^{-1}(y)$. A labeled example $(x, y)$ is classified correctly if $N(x) \subseteq h^{-1}(y)$. Following Cullina et al. [23], we define degraded hypotheses $\tilde{h} : \mathcal{X} \to \{-1, 1, \perp\}$,

$$\tilde{h}(x) = \begin{cases} y & : N(x) \subseteq h^{-1}(y) \\ \perp & : \text{otherwise.} \end{cases}$$

This allows us to express the adversarial classification accuracy of $h$, $1 - L(N, h, P)$, as

$$\frac{1}{2}(\mathbb{E}[\mathbf{1}[\tilde{h}(X_1) = 1]] + \mathbb{E}[\mathbf{1}[\tilde{h}(X_{-1}) = -1]]).$$

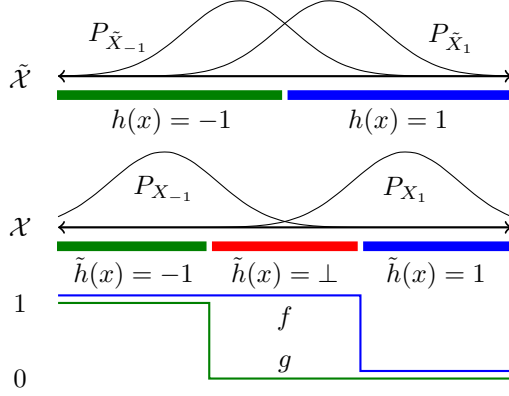

Figure 1: The relationships between a classifier $h : \mathcal{X} \to \{1, -1\}$, a degraded classifier $\tilde{h} : \tilde{\mathcal{X}} \to \{1, -1, \perp\}$, and potential functions $f, g : \mathcal{X} \to \mathbb{R}$.

Observe that $\mathbf{1}[\tilde{h}(x) = 1] + \mathbf{1}[\tilde{h}(x') = -1] \leq (c_N \circ c_N^\top)(x, x') + 1$. Thus the functions $f(x) = 1 - \mathbf{1}[\tilde{h}(x) = 1]$ and $g(x) = \mathbf{1}[\tilde{h}(x) = -1]$ are admissible potentials for $c_N \circ c_N^\top$. This is illustrated in Figure 1.

Our first theorem characterizes optimal adversarial robustness when $h$ is allowed to be any classifier.

**Theorem 1.** *Let $\mathcal{X}$ and $\tilde{\mathcal{X}}$ be Polish spaces and let $N : \mathcal{X} \to 2^{\tilde{\mathcal{X}}}$ be an upper-hemicontinuous neighborhood function such that $N(x)$ is nonempty and closed for all $x$. For any pair of distributions $P_{X_1}, P_{X_{-1}}$ on $\mathcal{X}$,*

$$(C_N \circ C_N^\top)(P_{X_1}, P_{X_{-1}}) = 1 - 2 \inf_h L(N, h, P)$$

*where $h : \tilde{\mathcal{X}} \to \{1, -1\}$ can be any measurable function. Furthermore there is some $h$ that achieves the infimum.*

In the case of finite spaces, this theorem is essentially equivalent to the König-Egerváry theorem on size of a maximum matching in a bipartite graph. The full proof is in Section A of the Supplementary.

If instead of all measurable functions, we consider $h \in \mathcal{H}$, a smaller hypothesis class, Theorem 1 provides a lower bound on $\inf_{h \in \mathcal{H}} L(N, h, P)$.

## 4 Gaussian data: Optimal loss

In this section, we consider the case when the data is generated from a mixture of two Gaussians with identical covariances and means that differ in sign. Directly applying (1) or (2), requires optimizing over either all classifiers or all transportation plans. However, a classifier and a coupling that achieve the same cost must both be optimal. We use this to show that optimizing over linear classifiers and 'translate and pair' transportation plans characterizes adversarial robustness in this case.

**Problem setup:** Consider a labeled example $(X, Y) \in \mathbb{R}^d \times \{-1, 1\}$ such that the example $X$ has a Gaussian conditional distribution, $X|(Y = y) \sim \mathcal{N}(y\mu, \Sigma)$, and $\Pr(Y = 1) = \Pr(Y = -1) = \frac{1}{2}$. Let $\mathcal{B} \subseteq \mathbb{R}^d$ be a closed, convex, absorbing, origin-symmetric set. The adversary is constrained to add perturbations to a data point $x$ contained within $\beta\mathcal{B}$, where $\beta$ is an adversarial budget parameter. That is, for all $x$, $N(x) = x + \beta\mathcal{B}$. This includes $\ell_p$-constrained adversaries as the special case $\mathcal{B} = \{z : \|z\|_p \leq 1\}$. For $N$ and $P$ of this form, we will determine $\inf_h L(N, P, h)$ where $h$ can be any measurable function.

We first define the following convex optimization problem in order to state Theorem 2. In the proof of Theorem 2, it will become clear how it arises.

**Definition 3.** *Let $\alpha^*(\beta, \mu)$ be the solution to the following convex optimization problem:*

$$(z, y, \alpha) \in \mathbb{R}^{d+d+1} \qquad \min \alpha \qquad s.t. \quad \|y\|_\Sigma \leq \alpha \qquad \|z\|_\mathcal{B} \leq \beta \qquad z + y = \mu \qquad (4)$$

*where we use the seminorms $\|y\|_\Sigma = \sqrt{y^\top \Sigma^{-1} y}$ and $\|z\|_\mathcal{B} = \inf\{\beta : z \in \beta\mathcal{B}\}$.*

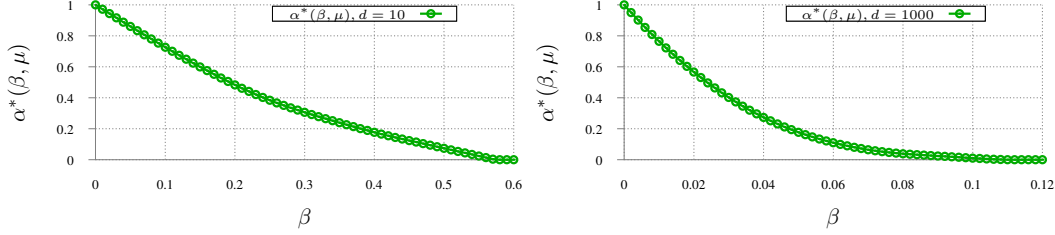

Figure 2: Variation in $\alpha^*$ w.r.t. $\beta$ for an $\ell_\infty$ adversary with $d = 10$ (**left**) and $d = 1000$ (**right**). $\alpha^*$ is the point at which the primal transport problem and the dual classification problem have matching solutions, given by $1 - 2Q(\alpha^*)$. The classification loss at this point is simply $Q(\alpha^*)$.

**Theorem 2.** *Let $N(x) = x + \beta\mathcal{B}$. Then $(C_N \circ C_N^\top)(\mathcal{N}(\mu, \Sigma), \mathcal{N}(-\mu, \Sigma)) = 1 - 2Q(\alpha^*(\beta, \mu))$, where $Q$ is the complementary cumulative distribution function for $\mathcal{N}(0, 1)$.*

The crucial properties of the solution to (4) are characterized in the following lemma.

**Lemma 1.** *Let $\mu \in \mathbb{R}^d$, $\beta \geq 0$, and $\alpha = \alpha^*(\beta, x)$. There are $y, z, w \in \mathbb{R}^d$ such that $y + z = \mu$ and*

$$\|y\|_\Sigma = \alpha \qquad \|z\|_\mathcal{B} = \beta \qquad \|w\|_{\Sigma*} = 1 \qquad \|w\|_{\mathcal{B}*} = \gamma \qquad w^\top y = \alpha \qquad w^\top z = \beta\gamma.$$

The proof of Lemma 1 is in Section B.1 of the Supplementary.

*Proof of Theorem 2.* We start from the definition of optimal transport cost and consider the restricted class of "translate and pair in place" couplings to get an upper bound. In these couplings, the adversarial attacks are translations by a constant: $\tilde{X}_1 = X_1 + z$ and $\tilde{X}_{-1} = X_{-1} - z$. The total variation coupling between $\tilde{X}_1$ and $\tilde{X}_{-1}$ does "pairing in place".

$$(C_N \circ C_N^\top)(P_{X_1}, P_{X_{-1}}) \leq \inf_{z \in \beta\mathcal{B}} C_{TV}(P_{\tilde{X}_1}, P_{\tilde{X}_{-1}}) = \inf_{z \in \beta\mathcal{B}} \sup_w 2Q\left(\frac{w^\top z - w^\top \mu}{\sqrt{w^\top \Sigma w}}\right) - 1.$$

The full computation of the total variation between Gaussians is in Section B.2 of the Supplementary.. The infimum is attained at $w^* = 2\Sigma^{-1}(z - \mu)$ and its value is $\sqrt{(z - \mu)^\top \Sigma^{-1}(z - \mu)}$. The choice of $z$ from Lemma 1 makes the upper bound $2Q(-\alpha^*(\beta, \mu)) - 1 = 1 - 2Q(\alpha^*(\beta, \mu))$.

Now we consider the lower bounds on optimal transport cost from linear classification functions of the form $f_w(x) = \text{sgn}(w^\top x)$. In the presence of an adversary, the classification problem becomes $\max_w \mathbb{P}_{(x,y)\sim P}[f_w(x + a_{w,y}(x)) = y]$. When $y = 1$, the correct classification event is $f_w(x + a_{w,1}(x)) = 1$, or equivalently $w^\top x - \beta\|w\|_{\mathcal{B}*} > 0$. This ultimately gives the lower bound

$$(C_N \circ C_N^\top)(P_{X_1}, P_{X_{-1}}) \geq \sup_w 1 - 2Q\left(\frac{\beta\|w\|_{\mathcal{B}*} - w^\top \mu}{\|w\|_{\Sigma*}}\right). \tag{5}$$

The full calculation appears in the supplementary material (Section B.3). From Lemma 1, there is a choice of $w$ that makes the bound in (5) equal to $1 - 2Q(\alpha^*(\beta, \mu))$. $\qquad\square$

The proof of Theorem 2 shows that linear classifiers are optimal for this problem. The choice of $w$ provided by Lemma 1 specifies the orientation of the optimal classifier.

## 4.1 Special cases

**Matching norms for data and adversary:** When $\mathcal{B}$ is the unit ball derived from $\Sigma$, the optimization problem (4) has a very simple solution: $\alpha^*(\beta, \mu) = \|\mu\|_\Sigma - \beta$, $y = \alpha\mu$, $z = \beta\mu$, and $w = \frac{1}{\|\mu\|_\Sigma}\Sigma^{-1}\mu$. Thus, the same classifier is optimal for all adversarial budgets. In general, $\alpha^*(0, \mu) = \|\mu\|_\Sigma$ and $\alpha^*(\|\mu\|_\mathcal{B}, \mu) = 0$, but $\alpha^*(\beta, \mu)$ can be nontrivially convex for $0 \leq \beta \leq \|\mu\|_\mathcal{B}$. When there is a difference between the two seminorms, the optimal modification is not proportional to $\mu$, which can be used by the adversary. The optimal classifier varies with the adversarial budget, so there is a trade-off between accuracy and robust accuracy.

$\ell_\infty$ **adversaries:** In Figure 2, we illustrate this phenomenon for an $\ell_\infty$ adversary. We plot $\alpha(\beta, \mu)$ for $\Sigma = I$ (so $\|\cdot\|_\Sigma = \|\cdot\|_2$) and taking $\mathcal{B}$ to be the $\ell_\infty$ unit ball (so $\|\cdot\|_\mathcal{B} = \|\cdot\|_\infty$). In this case (4) has an explicit solution. For each coordinate $z_i$, set $z_i = \min(\mu_i, \beta)$, which gives $y_i = \mu_i - \min(\mu_i, \beta)$, which makes the constraints tight. Thus, as $\beta$ increases, more components of $z$ equal those of $\mu$, reducing the marginal effect of an additional increase in $\beta$.

Due to the mismatch between the seminorms governing the data and adversary, the value of $\beta$ determines which features are useful for classification, since features less than $\beta$ can be completely erased. Without an adversary, all of these features would be potentially useful for classification, implying that human-imposed adversarial constraints, with their mismatch from the underlying geometry of the data distribution, lead to the presence of non-robust features that are nevertheless useful for classification. A similar observation was made in concurrent work by Ilyas et al. [39].

## 5 Gaussian data: Sample complexity lower bound

In this section, we use the characterization of the optimal loss in the Gaussian robust classification problem to establish the optimality of a rule for learning from a finite number of samples. This allows for precise characterization of sample complexity in the learning problem.

Consider the following Bayesian learning problem, which generalizes a problem considered by Schmidt et al. [68]. We start from the classification problem defined in Section 4. There, the choice of the classifier $h$ could directly depend on $\mu$ and $\Sigma$. Now we give $\mu$ the distribution $\mathcal{N}(\mathbf{0}, \frac{1}{m}I)$. A learner who knows this prior but not the value of $\mu$ is provided with $n$ i.i.d. labeled training examples samples. The learner selects any measurable classification function $\hat{h}_n : \mathbb{R}^d \to \{-1, 1\}$ by applying some learning algorithm to the training data with the goal of minimizing $\mathbb{E}[L(N, P, \hat{h}_n)]$.

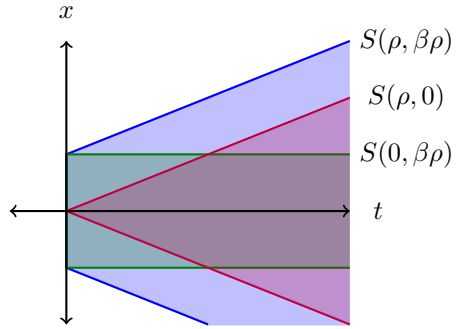

The optimal transport approach allows us to determine the exact optimal loss for this problem for each $n$ as well as the optimal learning algorithm. To characterize this loss, we need the following definitions. Let $\mathcal{A}$ be the $\ell_2$ unit ball: $\{y \in \mathbb{R}^d : \|y\|_2 \leq 1\}$. Let $S(\alpha, \beta) = \{(x, t) \in \mathbb{R}^d \times \mathbb{R} : x \in t\alpha\mathcal{A} + \beta\mathcal{B}\}$.

**Theorem 3.** *In the learning problem described above, the minimum loss of any learning rule is* $\Pr_{V \sim \mathcal{N}(\mathbf{0}, I)}[V \in S(\rho, \beta\rho)]$, *where* $\rho^2 = \frac{m(m+n)}{n}$.

The proof is in Section C of the Supplementary.

The special case where $\mathcal{B}$ is an $\ell_\infty$ ball was considered by Schmidt et al. [68]. They obtained a lower bound on loss that can be expressed in our notation as $\Pr[V \in S(0, \rho\beta)]$. This bound essentially ignores the random noise in the problem and computes the probability that after seeing $n$ training examples, the posterior distributions for $X_{n+1}|(Y_{n+1} = 1)$ and $X_{n+1}|(Y_{n+1} = -1)$ are adversarially indistinguishable. The true optimal loss takes into account the intermediate case in which these posterior distributions are difficult but not impossible to distinguish in the presence of an adversary.

Figure 3: $S(\rho, \beta\rho)$ is the set appearing in the statement of Theorem 3. $S(\rho, 0)$ corresponds to the loss lower bound obtained by Schmidt et al.. $S(0, \beta\rho)$ corresponds to the loss in the non-adversarial classification problem.

Schmidt et al. investigate sample complexity in the following parameter regime: $m = c_1 d^{\frac{1}{2}}$ which by design is a low noise regime. In this regime, they establish upper and lower bounds on sample complexity of learning an adversarially robust classifier: $C \frac{\beta^2 d}{\log d} \leq n \leq C' \beta^2 d$. By taking into account the effect of the random noise, our characterization of the loss loses this gap. For larger values of $m$, the difference between $\Pr[Y \in S(0, \rho\beta)]$ and $\Pr[Y \in S(\rho, \rho\beta)]$ becomes more significant, so our analysis is useful over a much broader range of parameters.

## 6 Experimental Results

In this section, we use Theorem 1 to find lower bounds on adversarial robustness for empirical datasets of interest. We also compare these bounds to the performance of robustly trained classifiers

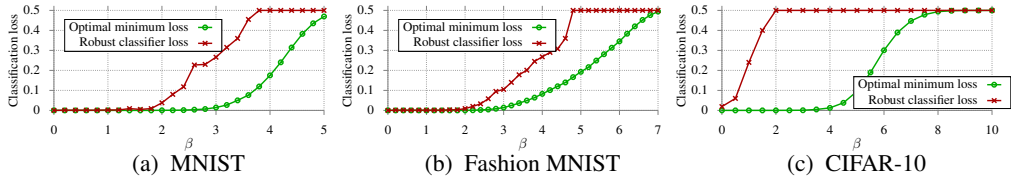

Figure 4: Variation in minimum $0 - 1$ loss (adversarial robustness) as $\beta$ is varied for '3 vs. 7'. For all datasets, the loss of a robustly classifier (trained with iterative adversarial training [54]) is also shown for a PGD adversary with an $\ell_2$ constraint.

on adversarial examples and find a gap for larger perturbation values. For reproducibility purposes, our code is available at `https://github.com/inspire-group/robustness-via-transport`.

## 6.1 Experimental Setup

We consider the adversarial classification problem on three widely used image datasets, namely MNIST [50], Fashion-MNIST [82] and CIFAR-10 [47], and obtain lower bounds on the adversarial robustness for any classifier for these datasets. For each dataset, we use data from classes 3 ($P_{X_1}$) and 7 ($P_{X_{-1}}$) to obtain a binary classification problem. This choice is arbitrary and similar results are obtained with other choices, which we omit for brevity. We use 2000 images from the *training set* of each class to compute the lower bound on adversarial robustness when the adversary is constrained using the $\ell_2$ norm. For the $\ell_\infty$ norm, these pairs of classes are very well separated, making the lower bounds less interesting (results in Section D of the Supplementary).

For the MNIST and Fashion MNIST dataset, we compare the lower bound with the performance of a 3-layer Convolutional Neural Network (CNN) that is robustly trained using iterative adversarial training [54] with the Adam optimizer [43] for 12 epochs. This network achieves 99.9% accuracy on the '3 vs. 7' binary classification task on both MNIST and Fashion-MNIST. For the CIFAR-10 dataset, we use a ResNet-18 [36] trained for 200 epochs, which achieves 97% accuracy on the binary classification task. To generate adversarial examples both during the training process and to test robustness, we use Projected Gradient Descent (PGD) with an $\ell_2$ constraint, random initialization and a minimum of 10 iterations. Since more powerful heuristic attacks may be possible against these robustly trained classifiers, the 'robust classifier loss' reported here is a lower bound.

## 6.2 Lower bounds on adversarial robustness for empirical distributions

Now, we describe the steps we follow to obtain a lower bound on adversarial robustness for empirical distributions through a direct application of Theorem 1. We first create a $k \times k$ matrix $D$ whose entries are $\|x_i - x_j\|_p$, where $k$ is the number of samples from each class and $p$ defines the norm. Now, we threshold these entries to obtain $D_{\text{thresh}}$, the matrix of adversarial costs $(c_N \circ c_N^\top)(x_i, x_j)$ (recall Section 3.2), whose $(i, j)^{\text{th}}$ entry is 1 if $D_{ij} > 2\beta$ and 0 otherwise, where $\beta$ is the constraint on the adversary. Finally, optimal coupling cost $(C_N \circ C_N^\top)(P_{X_1}, P_{X_{-1}})$ is computed by performing minimum weight matching over the bipartite graph defined by the cost matrix $D_{\text{thresh}}$ using the Linear Sum Assignment module from Scipy [41].

In Figure 4, we show the variation in the minimum possible $0 - 1$ loss (adversarial robustness) in the presence of an $\ell_2$ constrained adversary as the attack budget $\beta$ is increased. We compare this loss value to that of a robustly trained classifier [54] when the PGD attack is used (on the same data). Until a certain $\beta$ value, robust training converges and the model attains a non-trivial adversarial robustness value. Nevertheless, there is a gap between the empirically obtained and theoretically predicted minimum loss values. Further, after $\beta = 3.8$ (MNIST), $\beta = 4.8$ (Fashion MNIST) and $\beta = 1.5$, we observe that robust training is unable to converge. We believe this occurs as a large fraction of the data at that value of $\beta$ is close to the boundary when adversarially perturbed, making the classification problem very challenging.

We note that in order to reduce the classification accuracy to random for CIFAR-10, a much larger $\ell_2$ budget is needed compared to either MNIST or Fashion-MNIST, implying that the classes are better separated.

# 7 Related work and Concluding Remarks

We only discuss the closest related work that analyzes evasion attacks theoretically. Extensive recent surveys [10, 51, 64] provide a broader overview.

**Distribution-specific generalization analysis:** Schimdt et al. [68] studied the sample complexity of learning a mixture of Gaussians as well as Bernoulli distributed data in the presence of $\ell_\infty$-bounded adversaries, which we recover as a special case of our framework in 5. Gilmer et al. [33] and Diochnos et al. [26] analyzed the robustness of classifiers for specific distributions, i.e. points distributed on two concentric spheres and points on the Boolean hypercube respectively. In contrast to these papers, our framework applies for any binary classification problem as our lower bound applies to arbitrary distributions.

**Sample complexity in the PAC setting:** Cullina et al. [23], Yin et al. [85] and Montasser et al. [56] derive the sample complexity needed to PAC-learn a hypothesis class in the presence of an evasion adversary. These approaches do not provide an analysis of the optimal loss under a given distribution, but only of the number of samples needed to get $\epsilon$-close to it, i.e. to learn the best empirical hypothesis.

**Optimal transport for bounds on adversarial robustness:** Sinha et al. [73] constrain the adversary using a Wasserstein distance bound on the distribution that results from perturbing the benign distribution and study the sample complexity of SGD for minimizing the relaxed Lagrangian formulation of the learning problem with this constraint. In contrast, we use a cost function that characterizes sample-wise adversarial perturbation exactly, which aligns with current practice and provide a lower bound on the $0 - 1$ loss with an adversary, while Sinha et al. minimize an upper bound to perform robust training. Mahloujifar et al. [55] and Dohmatob [27] use the 'blowup' property exhibited by certain data distributions to provide bounds on adversarial risk, given some level of ordinary risk. In comparison, our assumptions on the example space, distribution, and adversarial constraints are much milder. Even in regimes where these frameworks are applicable, our approach provides two key advantages. First, our bounds explicitly concern the adversarial robustness of the optimal classifier, while theirs relate the adversarial robustness to the benign classification error of a classifier. Thus, our bounds can still be nontrivial even when there is a classifier with a benign classification error of zero, which is exactly the case in our MNIST experiments. Second, our bounds apply for any adversarial budget while theirs become non-trivial only when the adversarial budget exceeds a critical threshold depending on the properties of the space.

**Possibility of robust classification:** Bubeck et al. [13] show that there exist classification tasks in the statistical query model for which there is no efficient algorithm to learn robust classifiers. Tsipras et al. [79], Zhang et al. [87] and Suggala et al. [76] study the trade-offs between robustness and accuracy. We discuss this trade-off for Gaussian data in Section 4.

## 7.1 Concluding remarks

Our framework provides lower bounds on adversarial robustness through the use of optimal transport for binary classification problems, which we apply to empirical datasets of interest to analyze the performance of current defenses. In future work, we will extend our framework to the multi-class classification setting. As a special case, we also characterize the learning problem exactly in the case of Gaussian data and study the relationship between noise in the learning problem and adversarial perturbations. Recent work [30, 32] has established an empirical connection between these two noise regimes and an interesting direction would be to precisely characterize which type of noise dominates the learning process for a given adversarial budget. Another natural next step would be to consider distributions beyond the Gaussian to derive expressions for optimal adversarial robustness as well as the sample complexity of attaining it.

### Acknowledgements

We would like to thank Chawin Sitawarin for providing part of the code used in our experiments. This research was sponsored by the National Science Foundation under grants CNS-1553437, CNS1704105, CIF-1617286 and EARS-1642962, by Intel through the Intel Faculty Research Award, by the Office of Naval Research through the Young Investigator Program (YIP) Award, by the Army Research Office through the Young Investigator Program (YIP) Award and a Schmidt DataX Award. ANB would like to thank Siemens for supporting him through the FutureMakers Fellowship.

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
