[Supplementary Material · supp.pdf]

# Supplementary Material: Lower Bounds on Adversarial Robustness from Optimal Transport

**Arjun Nitin Bhagoji**
Department of Electrical Engineering
Princeton University
abhagoji@princeton.edu

**Daniel Cullina**
Department of Electrical Engineering
Pennsylvania State University
cullina@psu.edu

**Prateek Mittal**
Department of Electrical Engineering
Princeton University
pmittal@princeton.edu

## A   Proof of Theorem 1

First, we present an easy lemma that uses our topological conditions on the neighborhood function.

**Definition 1.** *A binary relation $R \subset \mathcal{X} \times \mathcal{Y}$, or equivalently a set-valued function $\mathcal{X} \to 2^{\mathcal{Y}}$, is upper hemicontinuous if it has the following property. For all open sets $V \subseteq \mathcal{Y}$ and points $x \in \mathcal{X}$ such that $R(x) \subseteq S$, $x$ has an open neighborhood $U$ such that $R(U) \subseteq V$. Equivalently, $R^T(\mathcal{Y} \setminus V)$ is closed.*

**Lemma 1.** *Suppose that the adversarial constraint function $N$ is upper hemicontinuous, and $N(x)$ is nonempty and closed for all $x \in \mathcal{X}$. Then the cost function $c_N \circ c_N^T$ is lower semicontinuous.*

*Proof.* For each point $(x, x')$ such that $(c_N \circ c_N^T)(x, x') = 1$, we will find an open neighborhood with the same cost. Thus $c_N \circ c_N^T$ is the indicator function of an open set and is lower semicontinuous.

The sets $N(x)$ and $N(x')$ must be disjoint because $(c_N \circ c_N^T)(x, x') = 1$. They are closed, and $\mathcal{X}'$ is a normal space, so they have disjoint open neighborhoods $V$ and $V'$. Because $N$ is upper hemicontinuous, $x$ and $x'$ have open neighborhoods $U$ and $U'$ such that $R(U) \subseteq V$ and $R(U') \subseteq V'$. Because $V$ and $V'$ are disjoint, $c_N \circ c_N^T$ is one everywhere in $U \times U'$.                               □

For the proof of Theorem 1 we need to use the concept of a cyclically monotone set [1].

**Definition 2.** *A subset $\Gamma \subseteq \mathcal{X} \times \mathcal{Y}$ is said to be c-cyclically monotone if, for all $n \in \mathbb{N}$ and all families of points $(x, y) \in \Gamma^n \subseteq \mathcal{X}^n \times \mathcal{Y}^n$,*

$$\sum_{i=0}^{n-1} c(x_i, y_i) \leq \sum_{i=0}^{n-1} c(x_i, y_{i+1})$$

*(with the convention $y_n = y_0$ ).*

*Proof of Theorem 1.* Abbreviate $c_N \circ c_N^T$ as $c$. From Lemma 1, the cost function $c$ is lower-semicontinuous. From Theorem 5.10 (ii), there is a set $\Gamma \subseteq \mathcal{X} \times \mathcal{X}$ that is measureable, is $c$-cyclically monotone, and such that every optimal coupling is concentrated on it.

We need to find $f, g : \mathcal{X} \to \mathbb{R}$ such that $c(x, y) \geq g(y) - f(x)$ everywhere and $c(x, y) \leq g(y) - f(x)$ for $(x, y) \in \Gamma$. The former property means that $f$ and $g$ are admissible potentials and the latter means that they are optimal in the dual transportation problem. A classifier $h$ can be constructed from any pair of admissible $\{0, 1\}$-valued potentials.

For all $i \geq 0$, let

$$
\begin{aligned}
A_0 &= \{x \in \mathcal{X} : \exists y' \in \mathcal{X} \text{ s.t. } c(x, y') = 1, (x, y') \in \Gamma\} \\
A'_{i+1} &= \{y' \in \mathcal{X} : \exists x \in A_i \text{ s.t. } c(x, y') = 0\} \\
A_{i+1} &= \{x \in \mathcal{X} : \exists y' \in A'_{i+1} \text{ s.t. } c(x, y') = 0, (x, y') \in \Gamma\} \\
B_0 &= \{y \in \mathcal{X} : \exists x' \in \mathcal{X} \text{ s.t. } c(x', y) = 1, (x', y) \in \Gamma\} \\
B'_{i+1} &= \{x' \in \mathcal{X} : \exists y \in B_i \text{ s.t. } c(x', y) = 0\} \\
B_{i+1} &= \{y \in \mathcal{X} : \exists x' \in B'_{i+1} \text{ s.t. } c(x', y) = 0, (x', y) \in \Gamma\}
\end{aligned}
$$

Further define $A = \cup_{i \geq 0} A_i$, $A' = \cup_{i \geq 1} A'_i$, $B = \cup_{i \geq 0} B_i$, and $B' = \cup_{i \geq 1} B'_i$. Observe that $A' = \{y \in \mathcal{X} : \exists x \in A \text{ s.t. } c(x, y) = 0\}$ and $B' = \{x \in \mathcal{X} : \exists y \in B \text{ s.t. } c(x, y) = 0\}$. If we let $g(y) = \mathbb{1}(B)$, then $f(x) = \mathbb{1}(B') = \sup_y g(y) - c(x, y)$, i.e. the largest function such that $g(y) - f(x) \leq c(x, y)$ everywhere. Alternative choices for $f$ and $g$ come from $A$ and $A'$. If we let $f(x) = 1 - \mathbb{1}(A)$, then $g(y) = 1 - \mathbb{1}(A') = \inf_x f(x) + c(x, y)$.

For all $x \in A$, there is some $j$ and sequences $(x_0, \cdots, x_{j-1})$ and $(y'_0, \cdots, y'_{j-1})$ such that $x_{j-1} = x$, $x_i \in A_i$, and $y'_{i+1} \in A'_{i+1}$ that witness this. Similarly, for all $y \in B$, there is some $k$ and sequences $(x'_0, \cdots, x'_{k-1})$ and $(y_0, \cdots, y_{k-1})$ such that $y_{k-1} = y$, $y_i \in B_i$, and $x'_i \in B'_i$. Now we have

$$
\sum_{i=0}^{j-1} c(x_i, y'_i) + \sum_{i=0}^{k-1} c(x'_i, y_i) = 2
$$

and

$$
\sum_{i=1}^{j-1} c(x_{i-1}, y'_i) + \sum_{i=1}^{k-1} c(x'_{i-1}, y_i) + c(x'_0, y'_0) + c(x_{j-1}, y_{k-1}) = c(x'_0, y'_0) + c(x_{j-1}, y_{k-1}).
$$

From the cyclic monotonicity of $\Gamma$ and the fact that $c$ is always at most 1, $c(x'_0, y'_0) = c(x_{j-1}, y_{k-1}) = 1$. Thus $c(x, y) = 1$ for all $(x, y) \in A \times B$. This means that $A$ and $B'$ are disjoint and $B$ and $A'$ are disjoint.

Now consider some $(x, y) \in \Gamma$. If $c(x, y) = 1$, then $x \in A_0$, $y \in B_0$, so $(x, y) \in A \times B$. If $c(x, y) = 0$, $(x, y)$ is in one of $A \times A'$, $B' \times B$, or $(\mathcal{X} \setminus A \setminus B') \times (\mathcal{X} \setminus A' \setminus B)$. We can now easily check that for $g(y) = \mathbb{1}(B)$ and $f(x) = \mathbb{1}(B')$, $g(y) - f(x) = c(x, y)$ everywhere in $\Gamma$. The choices $g(y) = \mathbb{1}(\mathcal{X} \setminus A')$ and $f(x) = \mathbb{1}(\mathcal{X} \setminus A)$ work similarly.

Finally, we have

$$
\begin{aligned}
&\mathbb{E}[g(X_{-1}) - f(X_1)] \\
&= \Pr[\tilde{h}(X_{-1}) = -1] - \Pr[\tilde{h}(X_1) \neq 1] \\
&= 1 - \Pr[\tilde{h}(X_1) \neq 1] - \Pr[\tilde{h}(X_{-1}) \neq -1] \\
&= 1 - \Pr[h(\tilde{X}_1) \neq 1] - \Pr[h(\tilde{X}_{-1}) \neq -1] \\
&= 1 - 2L(N, h, P).
\end{aligned}
$$

$\square$

# B  Full Proof of Theorem 2

For a closed convex ball $\mathcal{B} \subseteq \mathbb{R}^d$, define the cone $\mathcal{C_B} \subseteq \mathbb{R}^{d+1}$, $\mathcal{C_B} = \{(z, \alpha) : \alpha \geq 0, z \in \alpha\mathcal{B}\}$. Observe that $\mathcal{C_B}$ is convex and for $c \geq 0$, $(z, \alpha) \in \mathcal{C_B}$ implies $(cz, c\alpha) \in \mathcal{C_B}$. Thus $\mathcal{C_B}$ is indeed a cone. From this, define the norm $\|z\|_\mathcal{B} = \min\{\alpha : (z, \alpha) \in \mathcal{C_B}\}$. Thus $\mathcal{C_B} = \{(z, \alpha) : \|z\|_\mathcal{B} \leq \alpha\}$.

For a cone $\mathcal{C} \subseteq \mathbb{R}^d$, the definition of the dual cone is $\mathcal{C}^* = \{y \in \mathbb{R}^d : y^\top x \geq 0 \ \forall x \in \mathcal{C}\}$. A pair $(w, \gamma) \in \mathcal{C}_\mathcal{B}^*$ if and only if $w^\top z + \alpha\gamma \geq 0$ for all $(z, \alpha) \in \mathcal{C_B}$. It is enough to check the pairs $(z, \|z\|_\mathcal{B})$, which gives the condition $-w^\top z \leq \|z\|\gamma$.

This is very close to the ordinary definition of the dual norm. However, when $\mathcal{B}$ is not symmetric, the minus sign matters. If $\mathbf{0} \in \mathcal{B}$, then $(\mathbf{0}, 1) \in \mathcal{C}_\mathcal{B}$ and the constraint $\gamma \geq 0$ applies to $\mathcal{C}_\mathcal{B}^*$. However, if $\mathbf{0} \notin \mathcal{B}$, $\mathcal{C}_\mathcal{B}^*$ with contain points with negative $\gamma$ components. In this case, there is no interpretation as a norm.

## B.1    Proof of Lemma 1

Consider the following convex program:

$$(z, \alpha, y, \beta) \in \mathbb{R}^{d+1+d+1}$$
$$\min a\alpha + b\beta$$
$$(z, \alpha, y, \beta) \in \mathcal{C}_\mathcal{B} \times \mathcal{C}_\Sigma$$
$$z + y = \mu$$

The cone constraint is equivalent to $\|z\|_\mathcal{B} \leq \alpha$ and $\|y\|_\Sigma \leq \beta$. The equality condition is equivalent to $\mu - z - y \in \{\mathbf{0}\}$, the trivial cone.

The Lagrangian is

$$L = a\alpha + b\beta - w^\top(z + y - \mu)$$

$$= \begin{pmatrix} \mathbf{0}^\top & a & \mathbf{0}^\top & b \end{pmatrix} \begin{pmatrix} z \\ \alpha \\ y \\ \beta \end{pmatrix} - w^\top \begin{pmatrix} I & \mathbf{0} & I & \mathbf{0} \end{pmatrix} \begin{pmatrix} z \\ \alpha \\ y \\ \beta \end{pmatrix} + w^\top \mu$$

The dual is

$$w \in \mathbb{R}^d$$
$$\max \mu^T w$$
$$(-w, a, -w, b) \in \mathcal{C}_\mathcal{B}^* \times \mathcal{C}_\Sigma^*$$

The cone constraint on $w$ is trivial because the dual of $\{\mathbf{0}\}$ is all of $\mathbb{R}^d$.

If we change the objective of the first program to use a hard constraint on $\alpha$ instead of including it in the objective, the new primal is

$$(z, \alpha, y, \beta) \in \mathbb{R}^{d+1+d+1}$$
$$\min b\beta$$
$$(z, \alpha, y, \beta) \in \mathcal{C}_\mathcal{B} \times \mathcal{C}_\Sigma$$
$$z + y = \mu$$
$$\alpha \leq \alpha'$$

the new Lagrangian is

$$L = b\beta - w^\top(z + y - \mu) - \eta(\alpha' - \alpha).$$

The new dual is

$$(w, \eta) \in \mathbb{R}^{d+1}$$
$$\max \mu^T w - \alpha' \eta$$
$$\eta \geq 0$$
$$(-w, \eta, -w, b) \in \mathcal{C}_\mathcal{B}^* \times \mathcal{C}_\Sigma^*.$$

Rewriting without any cone notation, combining $\alpha$ with $\alpha'$, and specializing to $b = 1$, we have

$$(z, y, \beta) \in \mathbb{R}^{d+d+1}$$
$$\min \beta$$
$$\|z\|_\mathcal{B} \leq \alpha$$
$$\|y\|_\Sigma \leq \beta$$
$$z + y = \mu$$

and

$$(w, \eta) \in \mathbb{R}^{d+1}$$
$$\max \mu^T w - \alpha\eta$$
$$\eta \geq 0$$
$$\|-w\|_{\mathcal{B}}^* \leq \eta$$
$$\|-w\|_{\Sigma}^* \leq 1$$

From complementary slackness we have $-w^\top z + \eta\alpha = 0$ and $-w^\top y + b\beta = 0$. From the constraints, we have $\|z\|_{\mathcal{B}} \leq \alpha$, $\|y\|_{\mathcal{B}} \leq \beta$, $\|-w\|_{\mathcal{B}}^* \leq \eta$, and $\|-w\|_{\Sigma}^* \leq b$. We have $w^\top z \leq \|w\|_{\mathcal{B}}^* \|z\|_{\mathcal{B}}$ and $w^\top y \leq \|w\|_{\Sigma}^* \|y\|_{\Sigma}$. Combining these, all six inequalities are actually equalities.

## B.2 Simplification of transportation problem

From Theorem 1,

$$C_N \circ C_N^\top(P_{X_1}, P_{X_{-1}}) \leq \inf_{z \in \beta\mathcal{B}} C_{TV}(\tilde{P}_{X_1}, \tilde{P}_{X_{-1}}), \tag{1}$$

$$= \inf_{z \in \beta\mathcal{B}} \sup_A \tilde{P}_{X_1}(A) - \tilde{P}_{X_{-1}}(A), \tag{2}$$

$$= \inf_{z \in \beta\mathcal{B}} \sup_w \mathbb{E}_{x \sim \mathcal{N}(\mu-z,\Sigma)}\left[\mathbf{1}(w^\intercal x > 0)\right] - \mathbb{E}_{x \sim \mathcal{N}(-\mu+z,\Sigma)}\left[\mathbf{1}(w^\intercal x > 0)\right] \tag{3}$$

$$= \inf_{z \in \beta\mathcal{B}} \sup_w Q\left(\frac{w^\intercal z - w^\intercal \mu}{\sqrt{w^\intercal \Sigma w}}\right) - Q\left(\frac{w^\intercal \mu - w^\intercal z}{\sqrt{w^\intercal \Sigma w}}\right), \tag{4}$$

$$= \inf_{z \in \beta\mathcal{B}} \sup_w 2Q\left(\frac{w^\intercal z - w^\intercal \mu}{\sqrt{w^\intercal \Sigma w}}\right) - 1. \tag{5}$$

As before, since the $Q$-function decreases monotonically, its supremum is obtained by finding $\inf_w \frac{w^\intercal z - w^\intercal \mu}{\sqrt{w^\intercal \Sigma w}}$. The infimum is attained at $w^* = 2\Sigma^{-1}(z - \mu)$ and its value is $\sqrt{(z-\mu)^\intercal \Sigma^{-1}(z-\mu)}$, which implies that

$$C_N \circ C_N^\top(P_{X_1}, P_{X_{-1}}) \leq \inf_{z \in \beta\mathcal{B}} 2Q\left(\sqrt{(z-\mu)^\intercal \Sigma^{-1}(z-\mu)}\right) - 1. \tag{6}$$

## B.3 Connection to the classification problem

We consider the linear classification function $f_w(x) = \mathrm{sgn}(w^\intercal x)$.

**Classification accuracy:** We define the classification problem with respect to the classification accuracy $\mathbb{E}_{(x,y)\sim P}\left[\mathbf{1}(f_w(x) = y)\right] = \mathbb{P}_{(x,y)\sim P}\left[f_w(x) = y\right]$, which also equals the standard $0-1$ loss subtracted from 1. The aim of the learner is to maximize the classification accuracy, i.e. the classification problem is to find $w^*$ which is the solution of $\max_w \mathbb{P}_{(x,y)\sim P}\left[f_w(x) = y\right]$.

**Performance with adversary:** In the presence of an adversary, the classification problem becomes

$$\max_w \mathbb{P}_{(x,y)\sim P}\left[f_w(x + h(x, y, w)) = y\right]$$
$$= \max_w \frac{1}{2}\mathbb{P}_{x\sim \mathcal{N}(\mu,\Sigma)}\left[f_w(x + h(x, 1, w)) = 1\right] + \frac{1}{2}\mathbb{P}_{x\sim \mathcal{N}(-\mu,\Sigma)}\left[f_w(x + h(x, -1, w)) = -1\right].$$

We will focus on the case with $y = 1$ for ease of exposition since the analysis is identical. The correct classification event is then

$$f_w(x + h(x, 1, w)) = 1,$$
$$\Rightarrow w^\intercal(x + h(x, 1, w)) > 0,$$
$$\Rightarrow w^\intercal x - w^\intercal \operatorname*{argmax}_{z\in\beta\mathcal{B}} w^\intercal z > 0,$$
$$\Rightarrow w^\intercal x - \max_{z\in\beta\mathcal{B}} w^\intercal z > 0$$
$$\Rightarrow w^\intercal x - \beta\|w\|_* > 0,$$

Figure 1: Variation in minimum $0 - 1$ loss (adversarial robustness) as $\beta$ is varied for '3 vs. 7'. For MNIST and Fashion-MNIST, the loss of a robustly classifier (trained with iterative adversarial training) is also shown for a PGD adversary with an $\ell_\infty$ constraint.

where $\|\cdot\|_*$ is the dual norm for the norm associated with $\mathcal{B}$. This gives us the classification accuracy for the case with $y = 1$ as $\max_w \mathbb{E}_{x \sim \mathcal{N}(\mu, \Sigma)} \left[\mathbf{1}(w^\mathsf{T} x - \beta\|w\|_* > 0)\right]$. We now perform a few changes of variables to obtain an expression in terms of the standard normal distribution. For the first, we do $x' = x - \mu$, which gives us $\max_w \mathbb{E}_{x' \sim \mathcal{N}(\mathbf{0}, \Sigma)} \left[\mathbf{1}(w^\mathsf{T} x' + w^\mathsf{T}\mu - \beta\|w\|_* > 0)\right]$. The second is $x'' = w^\mathsf{T} x'$, which results in $\max_w \mathbb{E}_{x'' \sim \mathcal{N}(0, \sigma^2)} \left[\mathbf{1}(x'' + w^\mathsf{T}\mu - \beta\|w\|_* > 0)\right]$, where $\sigma = \sqrt{w^\mathsf{T}\Sigma w}$. Finally, we set $x''' = \frac{x''}{\sigma}$, leading to $\max_w \mathbb{E}_{x''' \sim \mathcal{N}(0,1)} \left[\mathbf{1}(x''' + \frac{w^\mathsf{T}\mu}{\sigma} - \frac{\beta\|w\|_*}{\sigma} > 0)\right]$. The classification problem is then

$$\max_w \frac{1}{2}\mathbb{P}_{x \sim \mathcal{N}(\mu, \Sigma)} \left[f_w(x + h(x, 1, w)) = 1\right] + \frac{1}{2}\mathbb{P}_{x \sim \mathcal{N}(-\mu, \Sigma)} \left[f_w(x + h(x, -1, w)) = -1\right],$$
(7)

$$= \max_w Q\left(\frac{\beta\|w\|_* - w^\mathsf{T}\mu}{\sqrt{w^\mathsf{T}\Sigma w}}\right).$$
(8)

Since $Q(\cdot)$ is a monotonically decreasing function, it achieves its maximum at $w^* = \min_w \frac{\beta\|w\|_* - w^\mathsf{T}\mu}{\sqrt{w^\mathsf{T}\Sigma w}}$. This is the dual problem to the one described in the previous section.

## C    Proof of Theorem 3

The proof of Theorem 3 is below. The assumptions and setup are in Section 5 of the main paper.

*Proof.* Let $\hat{\mu} = \mathbb{E}[\mu|((X_1, Y_1), \ldots, (X_n, Y_n)]$. A straightforward computation using Bayes rule shows that $X_{n+1} \cdot Y_{n+1}|((X_1, Y_1), \ldots, (X_n, Y_n)) \sim \mathcal{N}(\hat{\mu}, I)$. Thus after observing $n$ examples, the learner is faced with a hypothesis testing problem between two Gaussian distributions with known parameters. From Theorem 2, the optimal loss for this problem is $Q(\alpha^*(\beta, \hat{\mu}))$.

Furthermore, $\hat{\mu} = \frac{1}{m+n}\sum_{i=1}^{n} X_i$ and $\hat{\mu} \sim \mathcal{N}(\mathbf{0}, \frac{n}{m(m+n)}I)$. Averaging over the training examples, we see that the expected loss is

$$\mathbb{E}[Q(\alpha^*(\beta, \hat{\mu}))] = \Pr[T \geq \alpha^*(\beta, \hat{\mu})] = \Pr[(\hat{\mu}, T) \in S(1, \beta)] = \Pr[Y \in S(\rho, \rho\beta)]$$

where $T \in \mathbb{R}$, $T \sim \mathbb{N}(0, 1)$ and $V \in \mathbb{R}^{d+1}$, $V \sim \mathbb{N}(0, I)$. $\qquad\qquad\square$

## D    Results for an $\ell_\infty$ adversary

In Figures 1a and 1b, we see that the lower bound in the case of $\ell_\infty$ adversaries is not very informative for checking if a robust classifier has good adversarial robustness since the bound is almost always 0, except at $\beta = 0.5$, in which any two samples can be reached from one another with zero adversarial cost, reducing the maximum possible classification accuracy to 0.5. This implies that in the $\ell_\infty$ distance, these image datasets are very well separated even with an adversary and there exist good hypotheses $h$. For MNIST (till $\beta = 0.4$) and Fashion MNIST ($\beta = 0.3$), we find that iterative adversarial training is effective.

For the CIFAR-10 dataset 1c, non-zero adversarial robustness occurs after $\beta = 0.2$. However, current defense methods have only shown robust classification with $\beta$ up to 0.1, where the lower bound is 0. In future work, we will explore the limits of $\beta$ till which robust classification is possible with neural networks.

# References

[1] Cédric Villani. *Optimal transport: old and new*, volume 338. Springer Science & Business Media, 2008.