[Reviews · NeurIPS 2019]

Reviewer 1



The setting of binary classification with class-conditional data being Gaussians with identical covariance matrices gives interesting results but is too simple in terms of any novel technical challenges. Their setting ensures (needs a short proof) that optimizing over only linear classifiers and translate-and-pair transportation plans or couplings is sufficient to lower bound the adversarial robustness. It helps improve the clarity of their main message but brings the submission down a little bit on originality and significance metrics, in my opinion.

Reviewer 2



- It is worth mentioning that there has already being some theoretical work linking adversarial robustness to optimal transport. See "Generalized No Free Lunch Theorem for Adversarial Robustness", ICML 2019 - To be more rigorous, there are a few constraints which must be satisfied by f, g, and c in definition 2, for the ensuing arguments to be strictly valid. For example, f and g should be restricted to bounded continuous functions, etc. etc. - In the proof of Theorem 2, what is $\tilde{P}_{X_1}$ ? I checked over and over again, but this quantity was never defined in the manuscript. - In the proof of Theorem 2, what do you mean by "translate and pair in place" ? Please write out the form of such transportation plans explicitly. How do you get the upper bound in terms of TV distance ? Please provide all relevant details. - In inequality (1) of appendix B.2, what is \tilde{P}_{X_1} ? What is it's relation to minimization variable z ? How do you go from (2) to (3). The proof as it stands is at best not clear, and at worst incorrect. Too many underfined quantities, and unjustified transitions. - line 76: ... gives tilde{x} to the learner and ... - line 92: correctly, ==> correctly. - lines 224 -- 234: What is m ? m ==> d ? - line 177: missing verb "optimizing over plans us" ==> "optimizing over plans gives us" - In the equation just after line 196, the inf after the first inequality doesn't make sense (variable z is not used!) - What are X_1 and X_{-1} ? - Theorem 1 and the constructions leading to it could be greatly simplified by only considering "symmetric" neighborhoods N, i.e x' in N(x) iff x in N(x'). Also only considering spaces for which N(x) subseteq X for all x (which leads to tilde{X} = X), simplifies the whole thing, and is also a reasonable restriction since adversarial examples are only "genuine" if they are from the same space as clean ones - The Gaussian example considered in section 4 was already proposed in Tsipras'18 (ref. [74] ?). I hand-checked and the lower bound on adversarial error (see eq. 8 of supp. mat) proposed by the present authors matches what was obtained in that paper. The authors should check this, and perhaps add a paragraph in their paper making this point clear. These are all minor points, and should be easy to address. Major Edit: There is a Bug in proof of Theorem 1. ====================================== The condition under which Theorem 1 has been proven, namely the "separability condition", is too restrictive: it implies the joint distribution of $X$ and $Y$ is discrete! Indeed, I'll show that the set compliment $\mathcal Z\setminus (a_n)_n$ has zero measure, where $\mathcal Z = \mathcal X \times \{-1,1\}$. Indeed, $\mathcal Z \setminus (a_n)_n$ (assumed measurable!) contains no $a_n$, thus by the contrapositive of "separability condition", we must have $P(\mathcal Z \setminus (a_n)_n) = 0$. Thus the support of $P$ is a contained in the sequence $(a_n)_n$, and so the former must be a countable combination of atoms. In case my above measurability assumption (in brackets) is troublesome, one may remedy my argument as follows. Actually the point $\{a_n\}$ may be non-measurable. But we may denote by $c_n$ the infimum of measures of all measurable sets containing $a_n$, this infimum is realized (since the countable intersection of the sequence of minimizing sets is measurable itself), and the set $A_n$ which realizes it is an atom. In any case, the product spafce $\mathcal Z$ must be discrete! $\Box$ However (and fortunately), one can discard the troublesome "separability" condition and still prove that in Theorem 1, LHS >= RHS. This is anyways sufficient for obtaining lower bounds on universal error.

Reviewer 3



General comment: This paper provides a nice principled way to compute a theoretical lower bound for robust accuracy based on a distance metric between two classes in binary classification. This makes rigorous the intuition that the more separated (in the sense of the perturbation neighborhood the attacker can search in), the lower the optimal robust loss should be - bounded below by the optimal standard loss. Although I believe this type of result could have also been shown outside the optimal transport framework with perhaps less notation, it is one justified way to look at the problem. The paper is written in a coherent manner, the theory is clear as are the experiments. I am not very aware of similar work - if indeed it is the first to prove lower bounds for adversarial robustness using a distributional distance metric I would highly recommended this paper for publication at Neurips. Here are some questions that remain, from high to low level: - How could this framework be extended to multiclass classification? And how would the method scale with the number of classes? - For the CIFAR plots you should add robust classifier losses as well even though I understand they might not look that great, that is what we have at the moment! - For Figure 3, aparently there are training issues above certain beta. What is the adversarial training accuracy in those cases? I suspect that it is below 100%. In this case, choosing a sufficiently expressive model with the right parameters should mitigate this issue? - In proof of theorem 3: It should read Pr(V \in ...) on the RHS.

[Author Response · NeurIPS 2019]

We thank the reviewers for their insightful feedback, and we appreciate the opportunity to improve our paper. We will
address typos and notational inconsistencies in the updated version.

**Response to Reviewer 1**:

We would like to emphasize that Theorem 1 is the most important contribution of our paper due to its generality.
By considering the set of all possible classifiers, it provides lower bounds on adversarial robustness for *any pair of*
*class-conditional distributions*. As we show in our experimental results in Section 6, we are able to obtain lower bounds
for arbitrary real-world datasets by constructing the empirical distribution for these. In our estimation, these results
serve to provide theoretical validation for adversarial training for low perturbation budgets as well as to highlight the
gap to optimality for higher budgets.

Our focus on the Gaussian case, as a concrete application of the general theorem, is due to the attention this setting
has received in relevant previous work such as Schmidt et al. and our results provide a conclusive characterization of
the behavior of the optimal loss under different adversarial constraints. We show that the common assumption of the
optimality of a linear classifier even in the presence of an adversary is justified through a primal-dual equivalence.

In the Gaussian case, our sample complexity result follows directly from the expression for the optimal loss. In the
updated version, we will add experiments with synthetic data which validates this result empirically using standard
learning algorithms.

**Response to Reviewer 2**: We thank the reviewer for pointing us to Dohmatob's "Generalized No Free Lunch Theorem
for Adversarial Robustness" from ICML 2019. There are several key differences between the results as well as methods
in the two papers. We require very mild assumptions on the example space, distribution, and adversarial constraints
while the assumptions in Dohmatob's paper are more restrictive. Further, ours explicitly concern the adversarial risk
of the optimal classifier, while Dohmatob's relate adversarial and ordinary risks of a classifier. Thus, our bounds on
adversarial risk can still be nontrivial even when there is a classifier with an ordinary risk of zero, which is exactly
the case in our MNIST experiments. Finally, while Dohmatob's bounds become non-trivial only when the adversarial
budget exceeds a critical threshold depending on the properties of the space, ours apply for any adversarial budget.

We will add the explicit but mild conditions required on the example spaces, neighborhood relations, and potential
functions throughout Section 3. $X_1$ is a random example from class 1 and $X_{-1}$ is an example from class $-1$. We will
add back the explanation of this notation which we accidentally removed. $\tilde{P}_{X_1}$ should have been $P_{\tilde{X}_1}$ everywhere. We
will add a more explicit description of $\tilde{X}_1$ for the translate adversarial strategy. This makes the dependence on $z$ explicit
on line 196. We will also add a clearer description of the "translate and pair in place" coupling. Finally, in B.2., going
from (2) to (3) is a standard calculation for the total variation distance between Gaussians with the same covariance,
which we will add in the updated version.

There are at least a few interesting examples of adversaries that produce examples in a different space than the clean
examples, e.g. by erasing pixels in a image. Allowing symmetric nearness relations does not complicate the proofs, it
only requires us to keep track of the difference between $N(x)$ and $N^{-1}(\tilde{x})$.

While we were unable to find the same calculation for the upper bound on classification accuracy for the Gaussian case
in Tsipras et al. [74], we did find it in concurrent work from the same group (Ilyas et al., Arxiv: 1905.02175). We will
add a citation and comparison in the updated version.

**Response to Reviewer 3**: While other papers such as Sinha et al. [68] and Dohmatob use ideas from optimal transport,
we are the first to identify the precise distributional distance metric that provides tight lower bounds on adversarial
robustness. Comparisons with Sinha et al. are in Section 7 and we compare to Dohmatob above. We would like to
emphasize that our identification of this metric has allowed us to apply our theoretical results directly to practical
datasets of interest.

We are currently investigating the extension of our results to the multi-class case. There is a close connection between
our framework and targeted adversarial examples in the multi-class setting. In this case, the transportation distances
between all pairs of classes characterize the performance of an optimal classifier. Since the number of distances required
for this characterization scales as the square of the number of classes, we are attempting to understand how much
information is contained in the one-vs-rest distances. Exact characterization of classification accuracy with untargeted
adversarial examples seems to require higher order interactions between class distributions. However, usable bounds
using only pairwise distances are available, which we will demonstrate in follow-up work.

As the reviewer correctly notes, the robust classifier loss on CIFAR-10 is high even for small budgets. Nevertheless, we
will add these in the updated version of the paper. We also ran experiments with a $4\times$ larger model for MNIST, per the
reviewer's suggestion, and observed some mitigation of the robust training issues up to an $L_2$ budget of 4.6. We thank
the reviewer for pointing this out and we will update our plots with this model.

[Meta-Review · NeurIPS 2019]

The paper proposes a classifier-independent lower bound for binary classification in the adversarial setting. More precisely, Theorem 1 connects the "Bayes optimal" adversarial robustness error to a notion of separability, that is the transportation distance between the positive and negative points in the feature space, induced by moving points around according to the attack model (i.e the constraints on the attacker). The idea of making use of the Kantorovich-Rubinstein transportation distance (also known as Wasserstein distance) to increase robustness is in the air presently, this paper show how it can be used. It is interesting to also point out that the authors also show that their lower bound can be efficiently computed by convex optimization. The contribution is clearly related to learning theory, but also have interesting empirical validation. The paper is very well written and organised, containing conceptual examples related to multivariate Gaussians, for which concrete computations are done. Namely, they show how their approach compares against the robust accuracy of a adversarially trained model. I personally think that the results propose in this paper will have an impact for the ML community that are interested in robustness and adversarial attack prevention, I am recommending this paper for an acceptation.